# Risk of Infection and Duration of Protection after the Booster Dose of the Anti-SARS-CoV-2 Vaccine BNT162b2 among Healthcare Workers in a Large Teaching Hospital in Italy: Results of an Observational Study

**DOI:** 10.3390/vaccines11010025

**Published:** 2022-12-22

**Authors:** Domenico Pascucci, Adriano Grossi, Alberto Lontano, Eleonora Marziali, Mario Cesare Nurchis, Vincenzo Maria Grassi, Matteo Raponi, Giuseppe Vetrugno, Giovanni Capelli, Giovanna Elisa Calabrò, Domenico Staiti, Maurizio Sanguinetti, Gianfranco Damiani, Patrizia Laurenti

**Affiliations:** 1Fondazione Policlinico Universitario “A.Gemelli” IRCCS, 00168 Rome, Italy; 2Section of Hygiene, University Department of Life Sciences and Public Health, Università Cattolica del Sacro Cuore, 00168 Rome, Italy; 3Local Health Authority ASL Roma 1, 00193 Rome, Italy; 4Istituto Superiore di Sanità, 00161 Rome, Italy; 5Dipartimento di Scienze Umane, Sociali e della Salute, Istituto Superiore di Sanità, Università Degli Studi di Cassino e del Lazio Meridionale, 03043 Cassino, Italy

**Keywords:** vaccination, SARS-CoV-2, COVID-19, mRNA vaccine, booster, health personnel, immunity

## Abstract

The efficacy of the first schedule of COVID-19 mRNA vaccines has decreased after the surge of the Delta variant, posing the need to administer a booster dose to enhance the neutralising immune response. This study aims at evaluating the duration of protection given by the booster dose of Pfizer–BioNTech BNT162b2 mRNA vaccine in healthcare workers (HCWs) of a large teaching hospital in Rome and to analyse the factors associated with post-booster vaccination infections. Data about vaccinations of HCWs with the BNT162b2 vaccine and nasal swabs positive for SARS-CoV-2 were extracted from the digital archives of the hospital from 27 September 2021 to 31 May 2022. In total, 5770 HCWs were observed. The cumulative risk of becoming infected by SARS-CoV-2 increased with time (2.5% at 4 weeks, 17% at 12 weeks and 40% at 24 weeks) and was significantly higher for females, younger classes of patients and for those who had developed a hybrid immunity (natural infection plus one dose, namely the primary schedule, added to the booster dose) compared to those who had completed the three doses. This study describes the duration and the determinants of the protection against infections after the booster dose of COVID-19 vaccine, highlighting the need for continuous monitoring of vaccine-induced immunogenicity.

## 1. Introduction

From the start of the pandemic until early May 2022, almost 17 million SARS-CoV-2 infections were diagnosed in Italy, resulting in almost 167,000 deaths [1]. The national vaccination campaign against COVID-19 started on 27 December 2020 [2], and HCWs, people with an increased risk of serious illness and non-HCWs in essential services were the first to have access to the vaccination. Only later was the campaign extended to the entire general population, with a priority scheme based essentially on age [3].

As of early May 2022, in total, more than 136 million doses of COVID-19 vaccines had been administered in Italy, of which 47.3 million were first doses, almost 50 million second/unique doses and 39 million booster doses [1]. Of these, approximately 5 million doses were administered to HCWs, who received an mRNA vaccine in 95% of the cases.

The high efficacy of mRNA vaccines in reducing morbidity and mortality from COVID-19 has been demonstrated in both randomised clinical trials and observational studies on real-world scenarios [4,5,6,7]. In particular, the Pfizer–BioNTech BNT162b2 mRNA vaccine proved effective in preventing asymptomatic, symptomatic and severe SARS-CoV-2 infection [7,8,9,10]. A previous study of 6649 HCWs in our teaching hospital reported an efficacy of the primary schedule of BNT162b2 against SARS-CoV-2 infection of 91.5% [11]. Moreover, evidence in the scientific literature shows that providing the Pfizer–BioNTech BNT162b2 mRNA vaccine to HCWs results in significant economic benefits [12].

However, after extremely low infection rates had been recorded for several months, there was a resurgence of COVID-19 cases [13] from the spring of 2021 onwards, mainly caused by the arrival of the Delta variant: a decrease in the overall efficacy of the complete vaccine schedule with mRNA vaccines was observed, 4 weeks after administration of the second dose. In contrast, the vaccine’s efficacy in preventing severe COVID-19 infection remained relatively stable. However, around 6 months after the second dose, the vaccine’s efficacy against SARS-CoV-2 infection and against severe COVID-19 disease dropped to 33% for the Delta variant and 80% for the Alpha variant, respectively [3].

Therefore, evidence has shown the need to administer a booster dose after the primary vaccination schedule [3,14] and booster dose vaccination campaigns [15,16] have been undertaken in many countries, starting in September 2021, administering mRNA vaccines, as they are safer and more effective than viral vector vaccines [17,18].

With the circulars of 27 September and 8 October 2021, the Italian Ministry of Health started the administration of the “booster” dose with mRNA vaccines, at least 6 months after the completion of the primary schedule, in order to maintain or restore an adequate level of immune response [19,20]. The timeline was further shortened first to 5 and then to 4 months due to the arrival of the more prevalent Omicron variant [21]. The vaccination campaign initially targeted high-risk groups, either because of frail conditions that predispose to the development of serious disease, or because of occupational exposure; all other population groups were then gradually included [19,20,22,23].

The booster dose of mRNA vaccines was shown to induce both a qualitative and quantitative enhancement of the neutralising immune response compared to the two-dose primary course or compared to unvaccinated subjects [14,24,25]. Indeed, the booster dose proved to be 95% effective (95% CI, 90–98%) two months after its administration against symptomatic infection caused by the Delta variant. However, efficacy values dropped dramatically to around 40% with the arrival of the Omicron variant (BA.1 and BA.2) in the same time interval. Nevertheless, the efficacy against severe disease caused with Omicron remained high with efficacy values of around 90% [26]. Vaccination policies widely differed among the European countries, leading to various outcomes in terms of vaccination coverage [27,28].

In Italy, during the first phase of the pandemic, non-pharmaceutical interventions were introduced, consisting of both compulsory measures (isolation, quarantine, stay-at-home orders, prohibition of public gatherings, closure of non-essential activities, school closures) and voluntary measures (social distancing, handwashing, respiratory etiquette and universal mask use) [29]. The second half of 2021 saw an exclusive maintenance of voluntary measures, however, given the critical role played by HCWs, “in order to protect public health and maintain adequate safety conditions in the provision of care and assistance”, through Decree Law 44/2021, on 1 April 2021 the Italian government introduced, for this category, the mandatory requirement to undergo vaccination with a primary schedule to prevent SARS-CoV-2 infection. From 15 December 2021, with Decree Law 172/2021, the obligation was also extended to the booster dose. 

Moreover, five countries in Europe established mandatory vaccination for different population groups, in particular HCWs and workers in long-term care facilities [28]. These measures stipulate that the COVID-19 vaccination is an essential requirement for the exercise of the profession and the performance of work activities by health professionals and HCWs [30,31].

The aim of the present study is to evaluate the duration of protection given by the administration of the booster dose with Pfizer–BioNTech BNT162b2 mRNA vaccine with respect to the acquisition of infection in employees of a large teaching hospital in Rome and to analyse the factors associated with the occurrence of SARS-CoV-2 infection after the booster dose. 

## 2. Materials and Methods

### 2.1. Study Design and Participants

A retrospective cohort study was conducted among employees of Fondazione Policlinico Universitario “A. Gemelli” IRCCS Teaching Hospital to evaluate the duration of protection offered by a booster dose of the BNT162b2 vaccine during the second phase of the SARS-CoV-2 vaccination campaign. All employees were offered the vaccination at the Hospital Vaccination Centre of the Columbus Hub—Fondazione Policlinico Universitario “A. Gemelli” IRCCS.

Foundation employees (aged ≥18 years) who could provide written informed consent and those who had completed the primary course with BNT162b2 vaccine and received the booster dose with BNT162b2 vaccine at the Hospital Vaccination Centre were included. Also included were those employees who, having completed the vaccination schedule with BNT162b2 vaccine at an external vaccination centre, took the booster dose at the Hospital Vaccination Centre, providing the vaccination certificate for the first schedule showing the date, place, type of vaccine and batch of the vaccine.

Employees who received either the primary schedule or the booster dose, a vaccine other than BNT162b2 or those who received the booster vaccination with BNT162b2 but in a vaccination centre outside the foundation were excluded. Furthermore, employees who had taken the booster dose outside the study period, those who had already taken the second booster dose and those who had had a positive swab between the first and second dose and/or between primary and booster dose were not included.

### 2.2. Data Sources

The vaccination data were extracted from the digital archives of the Hospital Vaccination Centre for the period of interest, i.e., from 27 September 2021 to 31 May 2022. The time horizon was defined taking into account the circular of 14 September 2021, which opened up the possibility of administering the booster dose to immunocompromised users. The observation period was conventionally set to May 31^st^ 2022, considering that an average observation period of 6 months had elapsed for 95% of the participants.

A database was created with an identification code for each participant, which was unique and anonymous with an indication of professional category. Socio-demographic data (i.e., age and gender) were retrieved from the employees’ tax code. Information on the employment of staff (i.e., doctors, trainees, nurses, other HCWs, administrative staff) was obtained from the Human Resources Office of the hospital, while that concerning resident doctors was provided by the specialisation school secretariat of the Università Cattolica del Sacro Cuore.

The database catalogued the dates of the first vaccination schedule, the date of administration of the booster dose and any positive results of rapid antigenic swabs and/or PCR for SARS-CoV-2, considering the temporal collocation of these between the administrations (before the first dose, between the first and second dose, after the primary schedule, after the booster dose).

Data on positive swab results were extracted through the Health Surveillance Information Systems of the hospital. 

The hospital offered tests for SARS-CoV-2 detection to all staff working within the hospital. Asymptomatic HCWs were invited to undergo periodic checks initially with molecular tests and subsequently with the use of rapid antigenic tests as stipulated by a ministerial circular [32]. The frequency of voluntary testing was determined on the basis of occupational risk, the occurrence of COVID-19-like symptoms or close contact with a COVID-19-positive case. The PCR test was mainly offered to symptomatic personnel or for confirmation of doubtful antigenic test results.

A different occupational risk profile of SARS-CoV-2 infection was defined according to the operating unit to which each staff member belonged (see Table 1).

### 2.3. Statistical Analysis

Descriptive analysis was conducted using frequencies and proportions for categorical measures and median and interquartile range for continuous variables.

As the main outcome of interest, SARS-CoV-2 infection was considered, defined as positivity of a nose/oro-pharyngeal swab using RT-PCR or generation II or III antigenic methods.

The risk of infection was monitored in the period between 8 days after the third dose and the end of the observation period, 31 May 2022, and was estimated by means of the Nelson–Aalen cumulative risk function, stratifying for the main variables of interest (sex, age, occupational category, operating unit, infection before primary schedule (namely, hybrid immunity)). The log-rank test was used to detect any differences in the distribution of factors between those who did and did not develop the infection after the third dose. A *p*-value of less than 0.05 was considered statistically significant.

Cox regression was used to estimate the hazard ratio (HR) and the corresponding 95% confidence interval for infections after the third dose in relation to the characteristics under study. Covariates were selected based on relevance in the scientific literature (gender, age) and univariate analysis, selecting for the Cox model variables whose differences were found to have *p* < 0.1. The epidemic trend was taken into account in the construction of the model, stratifying the analyses by national transmission index (Rt), defined as the risk of community transmission in the Lazio region in the week of exposure to the third booster dose. For each covariate, verification of the proportional hazards assumption was performed using Schoenfeld residual tests.

Analyses were conducted with STATA 14 (StataCorp, Stata Statistical Software).

## 3. Results

Between 27 September 2021 and 31 May 2022, 8159 employees were observed. Of these, 6071 (74%) took the booster dose at the Hospital Vaccination Centre. Following the application of the exclusion criteria, the final cohort consisted of 5770 HCWs (Figure 1).

Of these participants, 60% were female, 60% were under 45 years of age and almost 40% of the subjects analysed were doctors or residents. The median age was 41 years (Table 2).

Six per cent of the HCWs were infected between 3 and 12 months prior to the administration of the first dose and completing the primary course, with a single inoculation, due to previous immunity (hybrid immunity) as per the ministerial circular of 7 July 2021 [33] and then took the booster dose. In our cohort, a total of 1994 cases (35%) of SARS-CoV-2 infection occurred after administration of the booster dose. 

The figures below show the overall cumulative hazard ratio and the stratified ones. The cumulative risk function, over the observation period, obtained by means of the Nelson–Aalen estimator shows a risk of becoming infected that increases with time. The cumulative risk function is 2.5% at 4 weeks, 17% at 12 weeks and 40% at 24 weeks (Figure 2).

Stratifying by sex, males show a cumulative risk of 15% at 12 weeks and 36% at 24 weeks. In females, this is higher at the respective weeks with values of 19% and 42%. The difference between the two curves in the log-rank test appears statistically significant (*p* < 0.001) (Figure 3).

Analysing the results of the stratification by age group, the risk of falling ill decreases with increasing age. Specifically, taking a follow-up time of 24 weeks as a reference, individuals in the 18–30 class had a risk of 48%, those in the 31–45 class of 45%, those in the 46–60 class of 46–60 %, while those over 60 had a risk of 26%, with a percentage reduction of 47% between the two groups. The log-rank test showed that the difference between the curves, in the respective age groups, was statistically significant (*p* < 0.001) (Figure 4).

Assessing the curves by occupational category, the risk is highest throughout the duration of the observations for the category nurses and other HCWs (43% and 42% at 24 weeks) and lowest for doctors (34% at 24 weeks), while resident doctors presented a risk of 24–40% and administrative staff of 37%. In the respective occupational categories, the log-rank test showed that the difference between the curves was statistically significant (*p* < 0.01) (Figure 5).

Stratifying by occupational risk profile of contagion, the Nelson–Aalen estimator reported similar estimates among the various groups. In particular, considering a follow-up time of 12 weeks as an example, the risk of infection was 0.17 for high-risk HCWs, 0.18 for those at moderate risk and 0.16 for those at low risk. This result was also confirmed by the log-rank test (*p* = 0.086) (Figure 6).

Comparing those who received two doses of vaccine as a primary course with those who received a hybrid primary course, i.e., infection followed by one dose of vaccine (hybrid immunity), it emerges that those with a hybrid immunisation had a lower risk of becoming infected after the booster dose. This difference starts to become clear after week 8. At 24 weeks, the cumulative risk was 40% in those with the three vaccine doses versus 26% in those with hybrid immunity, a percentage difference of 53%. The difference between the two curves in the log-rank test appears statistically significant (*p* < 0.001) (Figure 7).

Multivariate analysis by Cox regression (Table 3) showed that female individuals had a significantly higher risk of acquiring the infection than male individuals (HR = 1.12; 95% CI, 1.02–1.24). Statistically significant differences were also present for the age groups 31–45 (HR = 0.86; 95% CI, 0.76–0.98), 46–60 (HR = 0.63; 95% CI, 0.55–0.72) and >60 (HR = 0.50; 95% CI, 0.39–0.64) compared to the age group 18–30, confirming that the risk of acquiring the infection after the booster dose decreases with increasing age. No significant differences emerge with regard to the operating unit to which they belong. With respect to professional profile, no clear differences emerge between senior doctors, resident doctors, nurses and administrative staff, while a difference in risk bordering on significance can be found for other HCWs (*p* = 0.058). Analysing the role of hybrid immunity, on the other hand, a statistically significant reduction in the risk of acquiring the infection (HR = 0.59; 95% CI, 0.47–0.74) was noted in comparison with those who received the three doses without ever acquiring the infection.

## 4. Discussion

The results of the present study, based on real-world data, produced robust evidence regarding the incidence of breakthrough infections (post-vaccination infections) up to 6 months after administration of the booster dose of the BNT162b2 vaccine. Overall, the infection rate found in the analysed population reflects the epidemiological trend of the pandemic in the Lazio region, especially as a function of the spread of Omicron subvariants. One of the most surprising results of the present study is that protection provided by hybrid immunity is considerably more durable over time. Cox regression analysis, over the study period considered, reported that the risk is reduced by approximately 40% in individuals with hybrid immunity compared to those who took the three doses without becoming infected before the primary cycle. 

On the role of hybrid immunity, the study confirms what has already been reported in studies published in authoritative journals [34,35,36]. In particular, antibody dosage analyses showed that the neutralising potency and magnitude of the antibodies are further increased after an additional dose of vaccine, and this effect is even greater if the vaccine recipient has had a previous SARS-CoV-2 infection, resulting in 25- to 100-fold higher antibody responses, driven by memory B lymphocytes and CD4+ T lymphocytes, and broader cross-protection against variants [37,38,39]. With regard to efficacy, Hall et al. [35] studied 35,768 HCWs in England between December 2020 and September 2021 and observed that vaccination efficacy was higher and protection against Alpha and Delta variant infection was more durable over time in participants who received one or two doses of vaccine after a primary infection compared to those vaccinated with two doses but no previous infection. The study by Altarawneh et al. [40], on the other hand, focused on the Omicron BA.1 and BA.2 variants, stating that recent booster vaccination had moderate efficacy, whereas hybrid immunity from previous infection and recent booster vaccination conferred the strongest protection against infection, with an efficacy of approximately 80%. Ferrara et al., although on a much smaller sample, also analysed the nature of the time-varying effect of hybrid immunity, defined as two doses of mRNA vaccine and a previous natural infection, on the risk of re-infection after a booster dose. Cox proportional hazard regression analysis showed that the risk is reduced by approximately two-thirds in individuals with hybrid immunity. In addition, they observed that this effect began at around month three, showing that protection against infection decreased more rapidly in booster dose recipients without ever having contracted the infection [41]. Another interesting finding concerns the higher incidence of infection in females and younger people. This can be supported by the fact that the majority of HCWs, as also in our hospital, are women. In addition to this, there are a number of behavioral factors: younger people have more intense social relationships and higher rates of contact, and in the case of the female sex in particular [42] this appears even more true if we consider the role of mothers in contact with school-age children who are more likely to contract the disease [43]. There appear to be no significant differences between the different units (high, moderate, low risk). This evidence could be justified by the high prevalence of the infection at a national level and especially in the hospital context during the period under consideration such that no unit could be considered COVID-free. On the other hand, the data concerning the category of other HCWs, which mainly includes social workers (OSS) and auxiliary workers, are in line with what has already been reported in the literature. Modenese et al. in fact report that the risk of infection for this category is double that of doctors [44]. 

The research available to date was mainly based on data collected with shorter follow-up periods and in most cases before the various Omicron lineages emerged. In this sense, our results add important evidence on the comparison of protection conferred by the full vaccine cycle or by COVID-19 vaccines plus infection. This indicates that it may be time to change the current “one shoe fits all” approach by tailoring current vaccine schedules to the needs of a specific population with a view to personalisation. Knowing the timing of the decline of different immune responses, the strength and breadth of neutralising titres of a population or individual, as well as their T cell responses to SARS-CoV-2 variants, could give us insight into the appropriate timing of different administrations, stimulating the formulation of increasingly precise and effective ad hoc vaccine schedules.

The results obtained emphasise the importance of vaccination practice, especially in the hospital setting, in order to protect HCWs and patients and to ensure continuity of care. Moreover, in the light of the results obtained, one is encouraged to consider large-scale testing of mucosal vaccines that induce a stronger immunity at the virus entry port than intramuscular administration [45], which combined with classical vaccination could lead to a drastic reduction in infections and viral circulation. Therefore, a commitment at the macro and meso level of decision-makers to allocate resources to develop innovative and up-to-date vaccines and evidence-based vaccination programmes is called for. At the micro level, the various actors are encouraged to correctly apply the directives issued by the relevant bodies and to improve communication strategies, based on a solid preventive culture, in order to maintain a high level of attention towards COVID-19, especially among HCWs, also ensuring their vaccination at the hospital.

### Strengths and Limitations

The results of the present study must be considered in light of its weaknesses and strengths. In relation to the assessment of the duration of vaccine protection, an important limitation is that our results are not generalisable to the entire population, as HCWs have a higher risk of infection. However, the sample of HCWs considered is rather large, allowing us to strengthen the statistical validity of the results. The follow-up period appears to be uneven among the participants, however, approximately 61% of those vaccinated had received the vaccine by November 2021 and 95% by the end of 2021, so for most participants the period analysed was 5–6 months. A further limitation was the lack of stratification by individuals with pre-existing medical conditions that could affect vaccine efficacy. However, the relatively young age of the individuals in the sample suggests that the risk of having a pre-existing medical condition that could aggravate a possible SARS-CoV-2 infection is very low [46,47]. 

Another limitation is that, initially, the PCR molecular test was employed but, as a circular was released by the Italian Ministry of Health in February 2021, the use of second and third generation antigenic tests was approved for COVID-19 diagnosis as, in light of the high prevalence of the infection, they showed a high accuracy [32].

Furthermore, given the widespread distribution of the swabs, several employees did not undergo the latter at the Hospital Vaccination Centre, which nevertheless always systematically and periodically collected data on positive subjects even when the diagnostic test was carried out outside the hospital. Finally, the stratification by occupational risk certainly added value to the study although, given the high organisational complexity of our university hospital, for several operators there may have been rotation over the months between the different care settings. Finally, a further limitation could be related to the type of study design. Indeed, we did not consider all the possible confounding factors and biases that may influence vaccine efficacy and the occurrence of breakthrough infections. However, we included those primarily associated with outcome, and the use of robust statistical analysis allowed us to balance the factors potentially related to these confounders.

Further studies are needed to better understand the role of hybrid immunity and of infections acquired following the different vaccine doses on a large scale and to examine the risk of complications of natural infection such as long COVID syndrome. Furthermore, it would be useful to investigate the role of additional variables (e.g., immune system disorders, cigarette smoking, etc.) on the duration of vaccination-induced protection.

## 5. Conclusions

This research adds important real-world data to the SARS-CoV-2 vaccination campaign and allowed us to determine the duration of protection against breakthrough infections conferred by the vaccination for a period of 6 months after the booster dose. The data collected confirm the need for continuous monitoring of vaccine-induced immunogenicity and for establishing the exact timing of vaccination with precise and more effective vaccination schedules. With a view to the sustainability of the National Health Service, it is essential to maintain a high level of attention on these issues in order to stem the spread of COVID-19, especially in vulnerable settings, such as hospitals, where HCWs play a fundamental role for the entire community.

## Figures and Tables

**Figure 1 vaccines-11-00025-f001:**
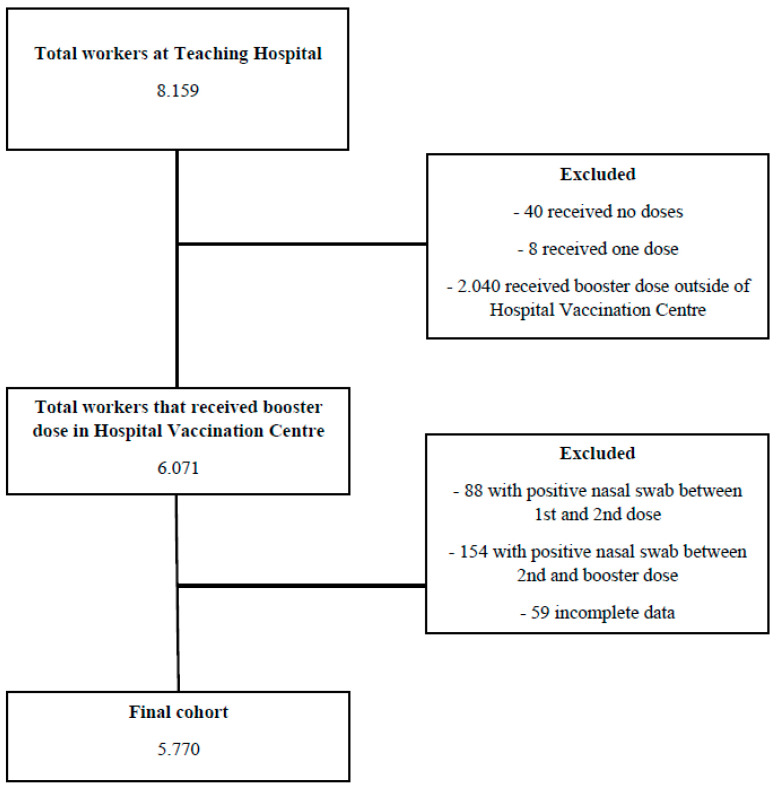
Flow-chart to identify the final cohort.

**Figure 2 vaccines-11-00025-f002:**
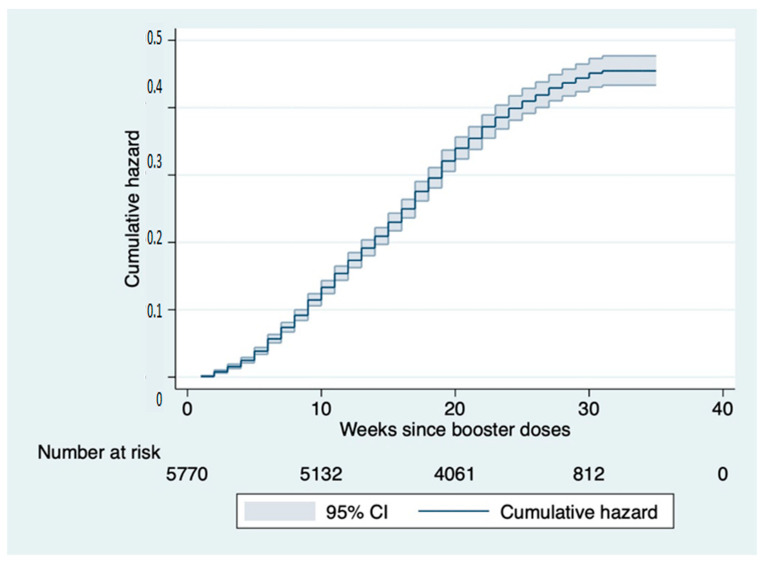
Nelson–Aalen cumulative hazard estimates: curve representing the cumulative risk of infection after booster dose in the study cohort over the follow-up period.

**Figure 3 vaccines-11-00025-f003:**
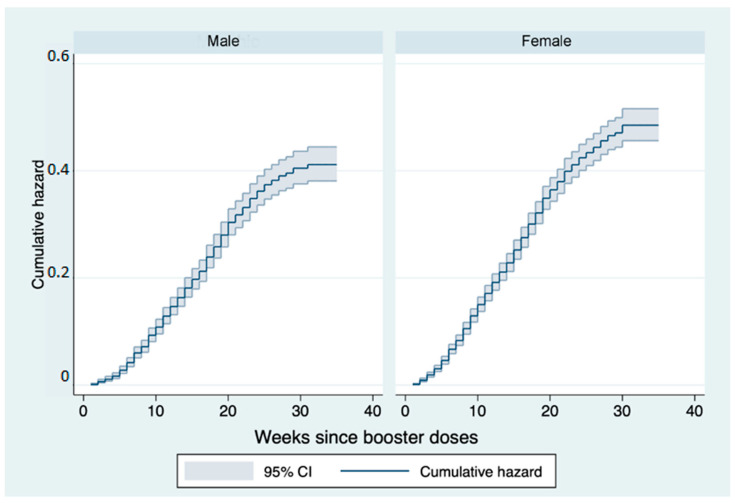
Nelson–Aalen cumulative hazard estimates: curves representing the cumulative risk of infection after booster dose stratified by sex in the follow-up period.

**Figure 4 vaccines-11-00025-f004:**
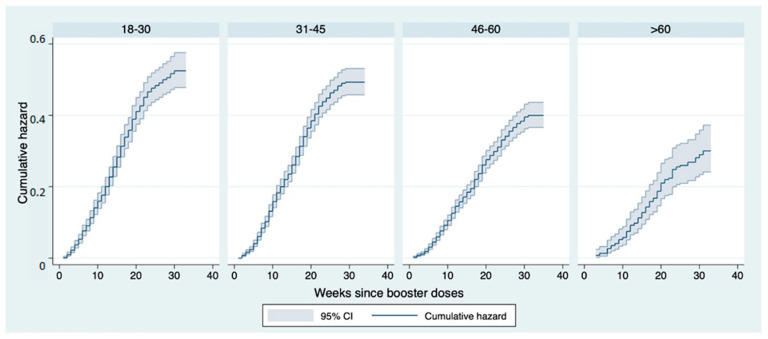
Nelson–Aalen cumulative hazard estimates: curves representing the cumulative risk of infection after booster dose stratified by age group in the follow-up period.

**Figure 5 vaccines-11-00025-f005:**
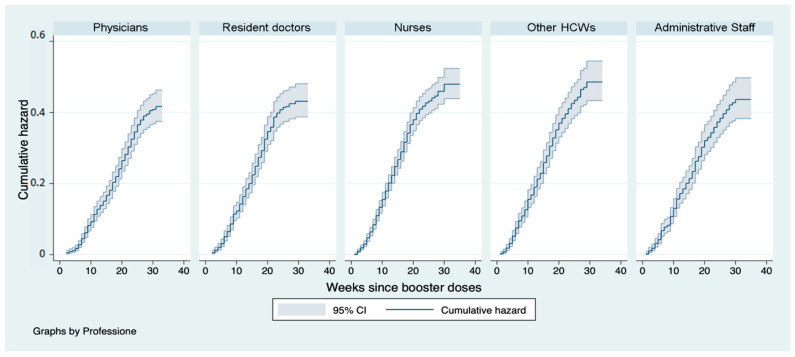
Nelson–Aalen cumulative hazard estimates: curves representing the cumulative risk of infection after booster dose stratified by occupational category in the follow-up period.

**Figure 6 vaccines-11-00025-f006:**
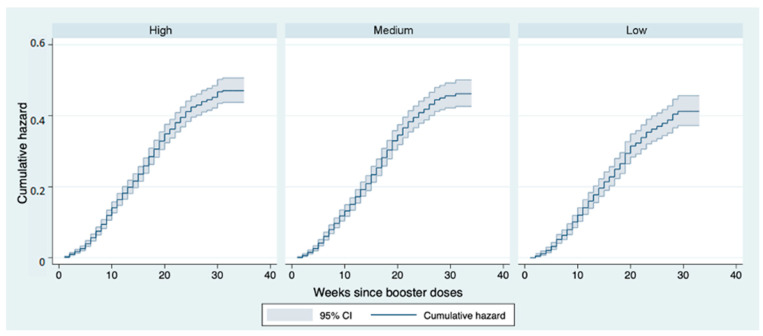
Nelson–Aalen cumulative hazard estimates: curves representing the cumulative risk of infection after booster dose stratified by operating unit in the follow-up period.

**Figure 7 vaccines-11-00025-f007:**
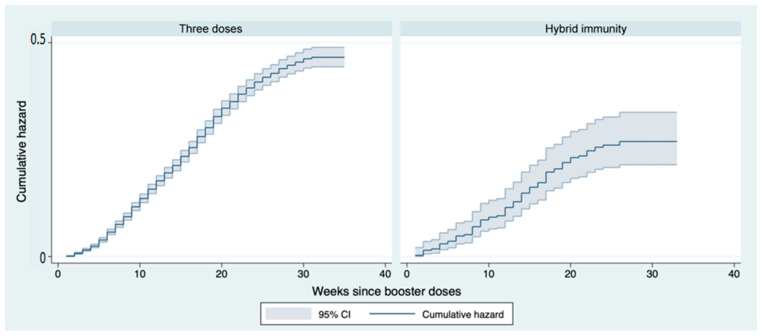
Nelson–Aalen cumulative hazard estimates: curves representing the cumulative risk of infection after booster dose stratified by type of immunity in the follow-up period.

**Table 1 vaccines-11-00025-t001:** Occupational risk profile, corresponding departments and frequency of collection of swabs for SARS-CoV-2.

Occupational Risk	Departments	Frequency of Collection of Swabs for SARS-CoV-2 (Days)
**High**	Emergency room sectors (general, pediatric, obstetrics); Organisational units; services and operative units (emergency department, radiodiagnostics, surgical digestive endoscopy service, operating theatres, resuscitation, intensive care, emergency medicine, emergency surgery, track units, etc.); COVID in-patient units.	14
**Moderate**	All the in-patient units and the staff working in them (haematology, radiotherapy, gynecological oncology, general liver transplant surgery, child neuropsychiatry, digestive surgery endocrine surgery).	21
**Low**	Paying wards; outpatient clinics; DH; services not included in the previous points; administrative functions.	35

**Table 2 vaccines-11-00025-t002:** Study population.

Variables	N (%)
**Age**	42 (21) *
**Sex**	
M	2307 (40)
F	3463 (60)
**Age class**	
18–30	1404 (25)
31–45	1976 (34)
46–60	2017 (35)
>60	373 (6)
**Professional category**	
Physicians	1151 (20)
Resident doctors	1045 (18)
Nurses	1717 (30)
Other HCWs **	1018 (18)
Administrative staff	839 (14)
**Occupational risk**	
High	2408 (42)
Moderate	2003 (35)
Low	1359 (23)
**Infection**	
Before primary schedule (hybrid immunity)	326 (6)
After booster dose	1994 (35)

* Median and interquartile range. ** The category “Other HCWs” includes midwives, healthcare assistants, pharmacists, psychologists, laboratory technicians, radiology technicians, physiotherapists, speech therapists, perfusionists, neurophysiopathology technicians, biologists, environmental and occupational prevention technicians, dieticians, orthoptists, audiometrists, occupational therapists and neuro- and psychomotricity therapists for children and adolescents.

**Table 3 vaccines-11-00025-t003:** Cox regression model: factors associated with SARS-CoV-2 infections after the booster dose.

	Haz. Ratio	P > z	IC 95%
**Sex**			
M	*Ref.*		
F	1.12	0.012	(1.02–1.24)
**Age class**			
18–30	*Ref.*		
31–45	0.86	0.034	(0.76–0.98)
46–60	0.63	0.001	(0.55–0.72)
>60	0.50	0.001	(0.39–0.64)
**Professional category**			
Physicians	*Ref.*		
Resident doctors	0.82	0.069	(0.68–1.02)
Nurses	1.08	0.283	(0.93–1.24)
Other HCWs	1.15	0.058	(0.99–1.34)
Administrative staff	1.16	0.118	(0.96–1.40)
**Occupational risk**			
High	*Ref.*		
Moderate	1.00	0.862	(0.90–1.12)
Low	0.91	0.235	(0.79–1.05)
**Immunity**			
3 doses	*Ref.*		
Hybrid	0.59	0.001	(0.47–0.74)

## Data Availability

The data presented in this study are available upon reasonable request from the corresponding author.

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
