# Peer review of "Risk of Infection and Duration of Protection after the Booster Dose of the Anti-SARS-CoV-2 Vaccine BNT162b2 among Healthcare Workers in a Large Teaching Hospital in Italy: Results of an Observational Study"

_vaccines, 2022, doi:10.3390/vaccines11010025_

Round 1

Reviewer 1 Report

General Comment:

The topic has been extensively dealt with in the literature from different aspects. Still, certainly, the amount of data analyzed here enriches and confirms other scientific work, adding some new elements that will need further research to be investigated. Most tables and figures need better formatting and aesthetics. In general, the level of scientific English is good.

Title: I suggest adding that the paper also searches the factors associated with post-booster vaccination infections and not only the “duration” of the protection. I also recommend adding the country “Italy” in the Title.  

Abstract: The abstract is complete and clear. It explains the aims of the paper well and includes all the necessary parts

Keyword: I suggest adding the keyword “Booster” and “ , and deleting “hospital” and “Teaching”

Introduction: This part is well written, but Authors can better explain the choices made in Italy on vaccination policies, highlighting the differences with other countries in the European Union or the world.

Material and Methods: This section is divided into sub-headings, which makes it easier to read. It’s good that it is reported that the Ethics Committee of the Hospital approved the study protocol.
There is no protocol adopted for all HCWs, and thus for some of them, the detection of infection over a large period of time was done with rapid antigenic testing, while for others, it was done with PCR testing. This may be a source of bias, which must be taken into account when analyzing the weaknesses of the paper in the discussion section.

Results: The flowchart is clear and help to understand the final cohort. The Cumulative Hazard ant the figures can be described better.

Table 1:  Need an overhaul in formatting, as I would suggest decreasing the size of the text and creating a different model for the table itself. Note that in the first Column the first row is empty.
The figures in this section are clear.
Table 2: In the Last row there is a mistake about Hybrid Immunity, 0-59 have to change in 0.59

Discussion: This section is well-written and clear. It is divided into subsections and it is essential that the limitations that have been taken into account are presented (we recommend adding the one explained above about the type of swab tests).

Reference: The number of references is good, but many of them do not refer to scientific works but to laws and protocols, diminishing the scientific basis of the work itself. It is recommended to increase the number of scientific paper by concentrating them on the introduction and materials and methods sections. There are some auto-references. 

Author Response

Point to point answers – Reviewer 1

Title: I suggest adding that the paper also searches the factors associated with post-booster vaccination infections and not only the “duration” of the protection. I also recommend adding the country “Italy” in the Title.  

We thank the reviewer for the remark. We modified the title accordingly.

Abstract: The abstract is complete and clear. It explains the aims of the paper well and includes all the necessary parts.

We thank the reviewer for the comment.

Keyword: I suggest adding the keyword “Booster” and “ , and deleting “hospital” and “Teaching”.

We thank the reviewer for the precious suggestion. We added the indicated keyword.

Introduction: This part is well written, but Authors can better explain the choices made in Italy on vaccination policies, highlighting the differences with other countries in the European Union or the world.

We thank the reviewer for the remark. We modified the text accordingly (lines 84-91 and lines 97-98).

Material and Methods: This section is divided into sub-headings, which makes it easier to read. It’s good that it is reported that the Ethics Committee of the Hospital approved the study protocol.
There is no protocol adopted for all HCWs, and thus for some of them, the detection of infection over a large period of time was done with rapid antigenic testing, while for others, it was done with PCR testing. This may be a source of bias, which must be taken into account when analyzing the weaknesses of the paper in the discussion section.

We thank the reviewer for the remark. A common protocol was adopted for all HCWs and is reported in lines 144-159.

Initally, the PCR molecular test was employed but, as the circular “Aggiornamento Sull’uso Dei Test Antigenici e Molecolari per La Rilevazione Di SARS-CoV-2” was released by the Italian Ministry of Health in February 2021, the use of second and third generation antigenic tests was approved for COVID-19 diagnosis as, in light of the high prevalence of the infection, they showed a high accuracy.

Anyhow this issue will be added in the limitations section (see lines 366-369).

Results: The flowchart is clear and help to understand the final cohort. The Cumulative Hazard ant the figures can be described better.

Table 1:  Need an overhaul in formatting, as I would suggest decreasing the size of the text and creating a different model for the table itself. Note that in the first Column the first row is empty.
The figures in this section are clear.
Table 2: In the Last row there is a mistake about Hybrid Immunity, 0-59 have to change in 0.59

We thank the reviewer for the remark.

We better clarified the differences among the attached figures (see lines 236-237, lines 245-246).

The editorial office of the journal is in charge of the formatting of tables. We corrected data in table 2 (now table 3), according to your suggestions.

Discussion: This section is well-written and clear. It is divided into subsections and it is essential that the limitations that have been taken into account are presented (we recommend adding the one explained above about the type of swab tests).

We thank the reviewer for the comment. We corrected the text accordingly.

Reference: The number of references is good, but many of them do not refer to scientific works but to laws and protocols, diminishing the scientific basis of the work itself. It is recommended to increase the number of scientific paper by concentrating them on the introduction and materials and methods sections. There are some auto-references. 

We thank the reviewer for the remark. We integrated the references accordingly.

Reviewer 2 Report

12Dec, 2022 Review for Vaccines of:

Duration of protection provided by the booster dose of the anti-2 SARS-CoV-2 vaccine BNT162b2 among healthcare workers in a 3 large Teaching Hospital in Rome: results of an observational 4 study

D. Pascucci, et al

General background and comments:

This s a well written manuscript by a group of distinguished clinicians and clinical investigators from Italy. The project is a chart- records retrospective evaluation of the longitudinal effectiveness of the vaccinee program specifically the infection outcomes related to the mRNA boosters. The data collected is reported to be de-identified and was reviewed and approved by the responsible Italian Human Studies Protection committee.

The population sampled was from their local medical center and includes all employees of the institution. Of their interest was the effectiveness of vaccine boosters within health care workers and defining specific groups that may have had differences in response to the booster.

Effectiveness was defined as new COVID infection and not resultant clinical outcomes such as hospitalization or death.

The overall project design and statistical treatment specifically are well designed and used validated population outcomes analytical tool.

Their primary study aim was to determine the clinical effectiveness to prevent COVID infection after the booster mandated by the Italian health authorities. The results largely track result in other countries that have access to the mRNA boosters.

The primary outcome is detection of mucosal infection and not the more dire adverse clinical outcomes of need for hospitalizations or worse.

The team has identified interesting outcomes which are novel and as such worth reporting. These include the gender differences, the potential differences in medical staff infection versus non-medical administrative and the most scientifically interesting- those that have had previous infections then boosted and the lower rate of incident infection (hybrid).

The claim of the “age’ phenomenon is somewhat exploratory as there are small number of subjects > 60 (6%).

The team has partially attempted to provide a scientific explanation for some of the findings, but it is incomplete.

From the reviewer’s perspective – the findings in this manuscript follow the know pathophysiology of SARS Co-V2 infection based on the characterized mutations within the SPIKE-RBD protein complex and behavior of clinically relevant viruses.

The manuscript overall would be improved by providing a rigorous context for the evolution of the virus which is well characterized in the literature. This would provide the reader a better understanding of how the described clinical outcomes could be supported based on how the corona virus has evolved over time reflected in infectively versus clinical virulence.

The manuscript is an excellent example of how the evolution of biology of the virus predicts clinical consequences and outcomes. The authors to their credit acknowledge that a “mucosal” focused vaccine holds potentially promise to eliminate the infection rate which is not a primary effect of the current parenteral approach.

It may be worthwhile to discuss briefly how the SARS-Cov 2 virus mutates within the RBD – binding domain to the ACE receptor as it is the basis for understanding OMICRON versus the earlier versions of the virus. Mutations within the RBD for OMICRON increase the ability to infect upper mucosal sites but not lower and as such have less risk for severe disease.

Specific comments- I have copied and pasted the lines from the text.

Line 77-80

“However, efficacy values dropped dramatically to around 40% with the arrival of the Omicron variant (BA.1 and BA.2) in the same time interval. Nevertheless, the efficacy against severe disease caused with Omicron remained high with efficacy values of around 90%.”

So, this validates the issue of mucosal infectivity versus more severe clinical consequences of the viral infection. The findings validate that parenteral administration will not significantly target direct mucosal immunity.

Currently the global focus is to reduce severe complications and death from COVID-19 and as boosters certainly have been reported to do this it would have been interesting to see the data related to the more severe outcomes in this cohort.

Lines 86-88

“…the obligation was also extended to the booster dose. These measures stipulate that the COVID-19 vaccination is an essential requirement for the exercise of the profession and the performance of work activities by health professionals and HCWs”

Th reviewer would like to know as I assume readers of the article what other measures were in place to prevent COVID? – Mandatory masking, limitations on interactions etc.  

Lines 90-94

“The aim of the present study is to evaluate the duration of protection given by the

administration of the booster dose with Pfizer-BioNTech BNT162b2 mRNA vaccine with

respect to the acquisition of infection in employees of a large Teaching Hospital in Rome

and to analyse the factors associated with the occurrence of SARS-CoV-2 infection after

the booster dose.”

As stated above, the vaccine boosters are not directed towards mucosal immunity- IgA etc and but they do promote innate immunity to prevent progression to severe disease hence their profound clinical benefit. But do these boosters in Italy continue to reduce the severe progression of COVID infections?  

Lines 103-106

“Foundation employees (aged ≥18 years) who could provide written informed consent

and those who had completed the primary course with BNT162b2 vaccine and received

the booster dose with BNT162b2 vaccine at the Hospital Vaccination Centre were included.”

Lines 119-120

“The vaccination data were extracted from the digital archives of the Hospital Vaccination Centre for the period of interest, i.e. from September 27th 2021 to May 31st 2022.”

The reviewer as a member of a US human studies protection review board, I have a question about these statements above. Usually a “waiver of consent” is granted in the US for chart review and if as stated the date is collected in a de-identified protocol the study is exempt. So, was all the data extracted from the EHR only for those that provided consent and what is current Italian regulations?

Table 1

“…those who carried out the booster vaccination with 111 BNT162b2 but in a vaccination centre outside the Foundation were excluded.”

It’s unfortunate that the team could not obtain validation of actual vaccination and the type of booster outside their institution as this is a significant proportion of the total. Is there a rationale why this was not pursued? Could the team have compared the outcomes in the outside cohort to those obtaining vaccination at their centre?

Line 146-159

“A different occupational risk profile of SARS-CoV-2 infection was defined according to the operating unit to which each staff member belonged. The 'high risk' band included the Emergency Room sectors (General, Paediatric and Obstetrics), but also the organisational units, Services and Operative Units necessary for their operation (e.g. Emergency Department, Radiodiagnostics, Surgical Digestive Endoscopy Service, Operating Theatres, Resuscitation, Intensive Care, Emergency Medicine, Emergency Surgery, ENT, TrackUnits, etc.) and the COVID Inpatient Units. For those in this band, swabs for SARS-CoV 2 were to be collected every 14 days. The 'moderate' risk band included all the in-patient   153 units in the NHS and the staff working in them (for example: haematology, radiotherapy, gynaecological oncology, general liver transplant surgery, child neuropsychiatry, digestive surgery, endocrine surgery) and the frequency of swabs was 21 days. The following were considered at 'medium-low' risk: paying wards, outpatient clinics, DH and services not included in the previous points, administrative functions. In this group, swabs were to be taken every 35 days.”

This is just a suggestion to the authors- This whole narrative could nicely be converted into a table for ease of reading and evaluation.

Defining high, moderate, and low risk in columns then the frequency of testing but adding the percentage of each complying with the mandates. Otherwise, it’s a bit difficult for the general reader to sort out

Summary

A well written and worthwhile study as reported in this manuscript. Although not revolutionary in the findings it is evolutionary clinically and but somewhat predictable based on the known mutational behavior of the Corona virus family.  There are somewhat novel findings that would be of interest to both clinicians and public health authorities.

1.     Subjects continue to acquire infection despite robust immune protection for progression to severe disease. This is somewhat concerning if a mutation develops that both increases infectivity and is less susceptible to the immune protection process there is potential for significant adverse health outcomes. Development of novel vaccines that target the mucosal boundary and possible targeting the non-mutation prone domain of the virus are needed

2.     The susceptibly based on age demonstrates the non-vaccine determinants of acquiring infection which certainly has public health and personal freedom implications.

3.     A lack of adverse outcomes based on pre-determine risk is interesting and suggests excellent risk mediation by the medical center to prevent infection

4.     Hybrid risk- another important finding on several levels. An argument from our patients is that once infected vaccine is not necessary. The findings refute his in fact support robust risk reduction when boostered and could serve as a strong public health message for vaccination.  The second implication is that infection with whole virus versus the RBD components of spike may clearly promote a superior immunological response with additional mucosal protection?

He authors have provided reasonable explanations for the findings and have addressed the limits of their study credibly.  Vaccines is an ideal journal for the manuscript.

I also believe this study is a tribute to the exceptional response in Italy to this horrific life changing event related to the pandemic. 
